# Radiography Managers’ Perspectives on the Strategies to Mitigate Disruptive Behaviours: A Qualitative Exploratory Study

**DOI:** 10.3390/healthcare10091742

**Published:** 2022-09-11

**Authors:** Bornface Chinene, Pauline Busisiwe Nkosi, Maureen Nokuthula Sibiya

**Affiliations:** 1Department of Radiography, Harare Institute of Technology, Harare 263, Zimbabwe; 2Faculty of Health Sciences, Durban University of Technology, 7 Ritson Rd., Musgrave, Berea, Durban 4001, South Africa; 3Division of Research, Innovation and Engagement, Mangosuthu University of Technology, Jacobs 4026, South Africa

**Keywords:** disruptive behaviour, mitigate, radiographers, patient safety, conflict resolution, emotional intelligence

## Abstract

Disruptive behaviours (DBs) are not normally in the scope of legal sanctions, notwithstanding their undesirable effects. Hence, many healthcare organizations still have difficulty in dealing with them in an effective manner. Several studies suggest that few organizations have tailor-made policies or procedures for evaluating, proving and mitigating these behaviours. However, evidence shows that mitigating DBs is critical to empowering healthcare workers to focus on providing superior, affordable and safe patient care. The aim of this study was to explore radiography managers’ perspectives on the strategies to mitigate DBs involving radiographers. An exploratory qualitative study employing one-on-one semi-structured in-depth interviews was carried out between March and April 2021. Eleven radiography managers at central hospitals in Harare Metropolitan Province were selected by criterion-purposive sampling. The interview data were analyzed using Tesch’s method of qualitative analysis. The data were first manually coded and then entered into Nvivo (QSR International Version 11) for further analysis. Three themes emerged from the interview data including awareness of DBs, willingness to address DBs, and conflict resolution. Context-specific strategies to mitigate DBs should be identified and implemented to guarantee a healthy work environment for radiographers so that they focus on providing excellent and safe patient care.

## 1. Introduction

Disruptive behaviours (DBs) are perceived by healthcare workers as predictors of adverse healthcare events [1]. The likelihood that these behaviours could compromise patient safety represents one of the major factors that led to their identification as a pressing issue in healthcare [2]. DBs are referred to as any conduct, whether verbal or physical, that negatively affects patient care. The majority of DBs that are reported involve bullying, harassment, disparaging statements, violent acts, and harsh language. Perpetrators of these behaviours include healthcare workers, administrators, patients or even patient escorts. DB is a concept that articulates human behaviour, work performance in healthcare and patient safety [3]. They are not normally in the scope of legal sanctions, notwithstanding their undesirable effects [4]. Hence, many healthcare organizations still have difficulty dealing with them in an effective manner [5]. DBs remain common occurrences, happening as frequently as daily in many healthcare settings [6]. However, evidence shows that mitigating DBs is critical to empowering healthcare workers to focus on providing superior, affordable and safe patient care [7].

Research findings in other healthcare professions and settings have shown that if not addressed, DBs can cause medical errors, staff resignations, wrong-site surgery, or even patient mortality [7,8,9]. The problem of DBs in radiography departments has been reported by many authors globally [10,11,12,13,14,15]. However, there is a dearth of literature that explores their mitigation. Addressing DBs involving radiographers is vital, as this professional group uses hazardous radiation in the execution of their duties, making patient safety of utmost importance.

Numerous studies highlight differences in DBs’ patterns, triggers and consequences amongst healthcare professionals or settings, as well as their effect on the selection of tools for effective mitigation [4,16]. A more practical method of mitigation and support by radiography leadership has been advocated by Willis, Friedman and Donnelly [17]. Promoting professional conduct and creating a culture of safety is difficult without leadership’s commitment, along with a framework for managing mitigation processes; suitable institutional policies; surveillance tools; training of healthcare team members; and accountabilities to each other [18,19]. The ethos of safety in an organisation starts with leadership because leadership drives ethos and ethos drive behaviour. Indeed, the next frontier in safety research is to assess and mitigate the human behavioural factors and processes that influence safety in the workplace [20]. Although in other healthcare settings, interventions have been developed to mitigate DBs and support victims [14], several studies suggest that few organizations have tailor-made policies or procedures for evaluating, proving, and mitigating these behaviours [21,22,23,24].

A recent study done to evaluate DBs involving radiographers at central hospitals in Harare reported that these behaviours are endemic. The most common behaviours were verbal abuse, sexual harassment and physical violence, respectively [25]. The same study determined that environmental or situational factors played a key role in eliciting such behaviours. The public healthcare work environment in Zimbabwe is characterised by recurrent labour disputes due to poor remuneration, shortage of equipment, deep-rooted cultural gender beliefs, outdated healthcare legislation and a style of management based on intimidation [26,27,28,29]. Studies have shown that the latter are predisposing factors to DBs in low-resource settings [30,31]. However, there is no policy to monitor and prevent these behaviours in the Zimbabwean radiography workforce. The Labour Act of Zimbabwe [32] makes provision for a fair and safe work environment. In addition, the Zimbabwe Patients’ Charter [33] and the Radiation Protection Act [34] espouse the rights of patients to safety during radiographic examinations. The aim of this study was to explore radiography managers’ (RMs) views on the strategies to mitigate DBs involving radiographers in central hospitals in Harare Metropolitan Province (HMP). It is anticipated that results from the study will provide a framework to address these unprofessional behaviours so that radiographers can focus on delivering superior and safe patient care.

## 2. Materials and Methods

An exploratory qualitative study design was followed between March and April 2021 employing one-on-one, face-to-face, semi-structured in-depth interviews with RMs. Since there are no explicit policies to mitigate DBs at central hospitals in HMP, this design allowed us to formulate a better understanding from the perspectives of RMs themselves. The participants were RMs from three central hospitals across five radiography departments in HMP who consented to take part in the study. A total of eleven RMs participated in the study. A further selection of the participants stopped after data saturation was reached, i.e., as more and more participants were interviewed, no new ideas or opinions emerged. The RMs were selected from across different hospitals and departments to ensure that the sample represented the whole province. Criterion-purposive sampling was employed to select the RMs, and the criterion was leadership. The RMs’ positions within the hospital would allow them to provide authoritative opinions about the mitigation of DBs in their respective departments. The RMs were informed of the purpose of the study and written informed consent was provided before participation.

As this study sought to explore different RMs’ viewpoints instead of seeking consensus, semi-structured one-on-one interviews were chosen over focus groups. Organising one-on-one interviews was also easier due to COVID-19 regulations. The researchers, referred to relevant literature [4,35], to develop the interview guide. Two lecturers at the Harare School of Radiography and one volunteer practising radiographer from each hospital were asked to review the interview guide. Demographic questions were introduced to build a profile of each interviewee and to add to the overall picture of RMs in HMP. The guide consisted of three open-ended questions that explored strategies to mitigate DBs: (1) What is your experience with DB complaints involving radiographers in your department? (2) Does your organisation have robust procedures and protocols for addressing DBs? If so, please tell me about these procedures; (3) In your opinion what strategies can be adopted to address/mitigate DBs at the: i. individual radiographer level, ii. radiography departmental level, iii. organizational level.

The interviews took place at a time and place proposed by the interviewee between 8 am and 4 pm, consistent with normal working hours for RMs in public service. Since the researchers and the participants had similar professional socialisation, this allowed the process of sharing ideas and work experiences. COVID-19 protocols according to the Ministry of Health and Child Care and World Health Organisation were observed. The interviews were audio-recoded by Rev Voice Recorder^TM^ for a duration of 45–60 min until data saturation.

The interviews were transcribed verbatim and entered in NVivo (Version 11) for analysis. NVivo is a qualitative data analysis (QDA) computer package produced by QSR international. The interview data were analysed, organised and interpreted using Tesch’s method of qualitative data analysis [36]. There are eight steps suggested for the method followed. These include:The interview transcripts were read and re-read to get an overall understanding of the whole interview.The shortest interview transcript was selected and after reading, thoughts about its general meaning were noted in the margins.All the transcripts were read and a list of topics was made. Similar topics were clustered together. The clustered topics were formed into major topics, unique topics, and leftovers.The list of topics that was created in the previous step was taken and the researcher went back to the data. The topics were shortened as codes and were written as codes next to the appropriate segments of the text. This was retried to see if new categories and codes emerged.The topics were then turned into categories. Related categories were then grouped together.A final conclusion was made on the number given to each category.After assembling the data fitting to each category, a preliminary analysis was done.The data were recorded again in an iterative fashion.

The significant statements with their meanings were extracted and analysed to formulate categories which improved the credibility of the interview data.

This study was assessed using the criteria for developing the trustworthiness of qualitative research as postulated by Lincoln and Guba [37], namely credibility, dependability, conformability and transferability. In the current study, credibility was achieved in various ways: Firstly, by the adoption of appropriate, well-recognised research methods, in this case, semi-structured in-depth interviews with RMs. Secondly, space triangulation was done. In this case, data were collected from RMs from five departments in three different hospitals in HMP. Thirdly, by allowing RMs to verify and edit the interview transcripts, the credibility of the study was further enhanced by the concept of “member checks” To attain dependability, an in-depth methodological description was made to allow the study to be repeated by anyone. To accomplish transferability, detailed background data were provided to establish the context of this study and a thorough description of data (thick description) so that readers could make contrasts to other settings based on as much information as possible.

## 3. Results

### 3.1. Demographics

The majority of the participants were in the 31–40-year old range (eight), two were above 40, and only one was in the 21–30 range. The work experience in years varied amongst the participants sampled. Participants with over 20 years of experience represented 10% of the sample size. Participants with 11–20 years represented 60%, and those with 10 years and under represented 30% of the sample size. Seven RMs had reached Masters level and there were four Bachelors, and only one had a Diploma. Table 1 summarizes the demographic characteristics of the participants.

### 3.2. Emerging Themes on Strategies to Mitigate DBs in HMP

This section reports three themes and ten subthemes that emerged from the interview data as summarized in Table 2 below. The three themes are as follows:Awareness of DBsWillingness to address DBs, andConflict resolution.


**Theme 1: Awareness of DBs**


This theme centres on the radiographers’ consciousness of DBs and their consequences on patient care and radiation safety. It includes subthemes of documentation, education/training, and emotional intelligence.

(i)Documentation

For DBs to be effectively mitigated, they should be documented so that specific measures to address them are formulated as explained by one of the participants below:


*“These behaviours are very common and from a policy point of view I believe that we need to document them so that we determine the prevalent ones and find specific measures to address them.”*

*(P#2; Female; PR)*


(ii)Education and training

After the cases have been documented, the majority of RMs suggests that the education and training of radiographers are important so that they are made aware of the impact of these behaviours on patient safety. However, one particular manager advocates for the academic curriculum to emphasize radiographer interaction with other health care workers and patients as noted below:


*“The truth is radiography is more than technical proficiency as you have to deal with other health care professionals and erratic patients. In the education and training, I think the part on psychology and patient management should emphasize a lot the interaction between other healthcare workers and patients. Students would then take the psychological and behavioural environment seriously.”*

*(P#2; Female; PR)*


Most managers agree that for them to effectively deal with DBs in their respective departments, they need extra training in human resources. Training should not be just limited to RMs but should extend to radiographers and other health care professionals as well. The quote below affirms this:


*“I think the first thing is, I am a radiographer. I have technical expertise as a manager. I might need an extra qualification that has to do with dealing with these behaviours or on human resources.”*

*(P#7; Female; CR)*


(iii)Emotional intelligence

Participants highlighted that radiographers and other healthcare workers need to develop emotional intelligence so that they can comprehend, use, and manage their own emotions in positive ways to relieve stress, communicate effectively, empathize with others, overcome challenges, and neutralize conflict in the workplace. This was expressed by one of the participants as indicated below:


*“… it takes us to the point of emotional intelligence. You need to understand why a certain problem has occurred. Maybe that person has got a genuine problem, and maybe that person just wants to be difficult.”*

*(P#4; Male; ACR)*


Another manager goes on to say:


*“The development of emotional intelligence amongst your staff is an important aspect even at home, people need to feel for each other. Empathy towards the patient, empathy towards other people, and appreciation of their existence.”*

*(P#8; Female; ACR)*



**Theme 2: Willingness to address DBs**


Willingness centres on the state of readiness to deliberately tackle the problem of DBs involving radiographers in HMP. It includes the categories of communication and taking charge.

(i)Communication

Proper communication was identified as one of the factors that could be used to mitigate DBs involving radiographers. The majority of RMs commented that radiographers, doctors, nurses, and patients must communicate efficiently and sufficiently with each other, to avoid barriers that would be created by miscommunication, leading to conflict. This is noted in the quotation below:


*“As a hospital, there should be improved interprofessional communication to avoid conflicts among healthcare workers.”*

*(P#1; Male; PR)*


Other RMs state that improved communication with patients in terms of regularly updating them about their waiting times or in case of emergencies that are going to need prompt attendance will avoid patients showing DBs. The excerpt below attests to this:


*“Communication for example is a good thing. You have a whole bench full of patients; just spare a moment to talk to them … communicate with them. So, if they are 50 of them they will know it will take an hour or so before they are attended to. So, inform them if there is a need for someone to skip the queue. Communicate with them constantly.”*

*(P#11; Male; CR)*


(ii)Taking charge

Most RMs reiterated that for DBs to be mitigated there needs to be a concerted and deliberate attempt from all the stakeholders involved, i.e., the individual radiographer, the radiography department, and the organisation as a whole. The following quotes exemplify this:


*“…. the individual should take charge of their health because this is a wellness issue. Their wellbeing is affected obviously, their work environment will be affected and everything else will be affected including patient care. So the individual themselves must take it up upon themselves to seek ways you know to address such challenges.”*

*(P#11; Male; CR)*



*“So, I think there should a holistic approach and deliberate attempt from top management to deal with these behaviours.”*

*(P#10; Male; CR)*



**Theme 3: Conflict resolution**


This is concerned with how leadership facilitates the peaceful ending of conflict and retribution, with both parties being satisfied that justice has been done. It includes subthemes: listen and understand, substitution, and external remedies.

(i)Listen and understand

Most RMs believe in solving the conflict harmoniously within the department first before reporting to structures outside the department. Structures outside the department are only sought if a solution is not found or the DB incident escalates as noted below:


*“… Personally, as a manager, I will try to sit down with the feuding parties and try to reach common ground. In the case where I do not succeed, I may have to take it up with my superiors…”*

*(P#1; Male; PR)*


RMs believe that listening and understanding from both feuding parties will help in solving the conflict. A conflict solved sets a precedence and hence instils confidence in the radiographers that if an incident is reported they will get justice. This is supported by the participants’ voices outlined below:


*“… there are times when you have to resolve conflict. Take people who are not working together well. Sit down with them and talk to them and make them understand why they are there and how a toxic work environment can affect the overall performance of the organisation. And when you resolve that conflict you realize that it won’t happen again.”*

*(P#10; Male; CR)*


Another RM suggests that despite human resources management being available in the organisation, they are supposed to be well equipped to deal with incidents of DBs in their immediate workplace. The selected quote below confirms this:


*“… we have HR that deals with these behaviours but as a manager you are also trained to deal with behaviours in your immediate workplace.”*

*(P#4; Male; ACR)*


Another RM is of the opinion that giving radiographers a platform to talk openly about their grievances and worries will help them vent their frustrations in a positive way so that they do not misdirect that anger to patients or other health care workers. One of the participants said:


*“Giving employees the platform to talk and let out what is inside of them. It will go a long way just to preserve the mental faculties of the radiographers.”*

*(P#9; Male; CR)*


(ii)Substitution

One of the ways of mitigating the consequences of DBs is to substitute the radiographer that has been involved in a DB incident. The resting of the radiographers enables them to recompose their faculties and ensures that they perform their duties when they are in the right frame of mind. The following excerpt below notes this:


*“I think that taking time off from the environment where the abuse occurred can really help. The victim is supposed to be given some days off to go and recover at home in peace.”*

*(P#1; Male; PR)*


Giving the radiographers some time off may act as an incentive to encourage reporting of such incidents as noted below:


*“As an incentive to encourage reporting we could offer off days to the victim. First we address the abuse and give some time off work.”*

*(P#1; Male; PR)*


(iii)External remedies

In cases where organisational remedies fail, for example in the case of very powerful specialists or if it involves very senior people in the organisation, then RMs suggest employing remedies that are from outside the organisational establishment. These may include professional boards. The excerpt below illustrates this:


*“Most managers will say if I address this issue head-on…I will lose business if this guy leaves. So maybe if it gets addressed from a professional board-level…”*

*(P# 10; Male; CR)*


If the incident involves harassment from a woman’s point of view, then pro-women groups may be employed. The passage below affirms this:


*“But the only way it can be dealt with is by seeking these pro-women organisations anonymously. Then these people will just come and tell the person that, no we have reports like this… maybe from an employee that has since resigned. At least if he is made aware that complaints are coming in, there will be that restraint on the individual.”*

*(P#11; Male; CR)*


## 4. Discussion

Disruptive behaviours involving radiographers have been shown to be rampant in HMP, and radiographers have been suffering from these perilous incidents. The reported behaviours are verbal abuse, sexual harassment and physical violence, respectively [25]. However, there is a dearth of literature that explores strategies to mitigate these behaviours in HMP. Furthermore, there are no tailor-made policies that attempt to combat these unprofessional behaviours within the radiography labour force. Disruptive behaviours are not normally in the scope of legal sanctions, notwithstanding their undesirable effects [4]. Healthcare organisations are given the power to deal with them, taking into account the settings’ unique circumstances [2]. This study represents one of the first studies seeking to identify context-specific strategies to mitigate DBs involving radiographers at central hospitals in HMP. It is anticipated that it will influence policy development aimed at finding interventions to address these unprofessional behaviours not only in radiography but also in Zimbabwean public healthcare. Without a doubt, the failure of administrations and scholars to provide policy-makers with evidence-based information is tantamount to a failure to address the problem [38,39]. Whereas this study focusses on the perspective of RMs, future studies involving all healthcare managers in HMP may be necessary to ascertain whether they have similar perceptions about DBs in their respective work settings.

The findings are taken to mean that radiographers and other healthcare workers should be made conscious of DBs in their workplaces. This is in agreement with other scholars, for instance, according to Rosenstein [5], awareness is the first strategy in mitigating DBs. A lot of people may not be mindful of their actions being seen as disruptive and are also unconscious of the possible negative subsequent results. Raising awareness and suggesting how they could have better managed the state of affairs will enable many of these people to correct themselves. Awareness is also vital because healthcare workers have frequently been subjected to these events such that they are often accepted as “part of the job” [35]. In addition, Grissinger [40] argues that responsibility for mitigating the problem belongs to the leaders, who need to raise awareness of the problem, inspire others to change, communicate respect as a core value, articulate their commitment to achieving it, and create a sense of urgency around doing so.

Nonetheless, Longo [22], asserts that when raising awareness, it is imperative to describe the types of behaviour that are considered disruptive, and these should be communicated. The descriptions are vital since they provide a context that enables leaders to effectively provide peer review of colleagues and also provide a framework for analysis. However, Vukmir [4] notes that there are significant regional and geographic variations in the behaviours described. This underscores the need to develop a definition of DBs that takes into consideration the values, culture and perceptions of the healthcare setting concerned [16,30]. By carrying out this study, insights on what constitutes DBs in the Zimbabwean healthcare settings were obtained.

One of the most important ways of making radiographers aware is by having education or training on how to deal with DBs. Most managers agree that for them to effectively deal with DBs in their respective departments, they need extra training in human resources. They argue that the radiography course focuses more on technical proficiency and the human aspect is usually overlooked. These findings are in harmony with the literature. Studies indicate that leaders might be unwilling to challenge persons who are disruptive because they might not have the knowledge to deal with the issue [4,19,22,41]. It is not a topic taught in training programs, so leaders may hesitate to take on a problem for which there is no obvious solution [18,42]. This, therefore, emphasises the need to educate or train both managers and radiographers in HMP on human resources issues.

Also, the findings note that radiographers need to develop emotional intelligence. With emotional intelligence, radiographers can comprehend, use, and manage their own emotions in positive ways to relieve stress, communicate effectively, empathize with others, overcome challenges and neutralize conflict in the workplace. Several studies have reported that the development of emotional intelligence in the workplace may lessen the occurrence of DBs [42,43,44] and this is also congruent with the findings of this study. However, more research is needed to determine if efforts on emotional intelligence will lead to a decline in incidents of DBs involving radiographers.

The findings indicate that for DBs to be mitigated, there should be a deliberate attempt to tackle the problem from all concerned stakeholders. One of the ways in which stakeholders may commit to dealing with DBs is by ensuring adequate and efficient communication between all the concerned parties. Proper communication was identified as one of the factors that can be used to mitigate DBs involving radiographers. The majority of RMs commented that radiographers, doctors, nurses, and patients must communicate efficiently and sufficiently with each other, to avoid barriers that would be created by miscommunication, leading to conflict. These findings are in harmony with the literature, for instance, Oliveira et al. [3] assert that poor communication can lead to personality conflicts, which provoke DB. To be precise, poor communication skills can influence other causes or triggers of DBs, and it is therefore considered as one of the most significant associated factors. A study that was done in China investigating hospital staff, although excluding radiographers, determined that 93.0% of DBs were related to inadequate communication between hospital staff and patients [45]. Another study by Lux et al. [46], exploring DBs involving nurses suggested educational strategies that concentrated on communication skills for professional practice.

Conflict resolution is concerned with how leadership facilitates the peaceful ending of conflict and retribution with both parties being satisfied that justice has been done. Findings reveal that RMs believe in solving the conflict harmoniously within the department first before reporting to structures outside the department. Radiography managers believe that listening and understanding from both feuding parties will help in solving the conflict. A conflict solved sets a precedent and hence instils confidence in the radiographers that if an incident is reported they will get justice. However, in cases where organisational remedies fail, for example in the case of very powerful specialists or if it involves very senior people in the organisation, then RMs suggest employing remedies that are from outside the organisational establishment, e.g., professional boards or pro-women organisations.

A number of studies in the literature underscore the importance of conflict resolution as a strategy to mitigate DBs in healthcare [17,40,47,48,49] which is similar to the findings of the current study. Johnson [50], asserts that ongoing DB must be addressed and this could take the form of conflict resolution. However, conflict resolution should be accompanied by team-building exercises to foster improved professional interactions. Vukmir [4], further affirms that conflict resolution is an efficient and effective way to mitigate DBs. However, the cornerstone of success in conflict resolution is to arrive at a mutually agreeable solution, not a one-sided approach to the problem. On the other hand, Grissinger [40] argues that an escalation policy must be established to manage conflicts when the standard communication process fails to resolve a dispute. Healthcare workers must know who to call to help in getting a fair resolution. Organisations should ensure that the conflict resolution process provides an avenue for resolution outside the typical chain of command in case the conflict involves a subordinate and his or her supervisor.

## 5. Limitations

Only RMs at central hospitals in HMP participated in the study due to time and resource limitations. As a result, these findings might not be generalized to Zimbabwe’s private healthcare facilities or to any other categories of public healthcare institutions. The major healthcare facilities in Zimbabwe are, however, the central hospitals, which, given their high patient loads and resource limitations, are particularly prone to DB incidents.

## 6. Conclusions

Certainly, the next frontier in safety research is to evaluate and address the human behavioural factors and processes that influence safety in the workplace. For radiographers, a healthy state of mind at work and patient safety is fundamental due to the use of radiation. Radiography managers have highlighted the strategies that are practical, relevant, and specific in their settings to address these unprofessional behaviours. Context-specific strategies to mitigate DBs must be implemented to guarantee a healthy work environment for radiographers so that they focus on providing excellent and safe patient care.

## Figures and Tables

**Table 1 healthcare-10-01742-t001:** Demographics of RMs in HMP.

Participant (P#) Number	Gender		Age (Years)			Educational Level			Position			Work Experience (Years)		
	M	F	21–30	31–40	>40	Dip.	BSc.	MSc.	Principal Radiographer (PR)	Ass. Chief Radiographer (ACR)	Chief Radiographer (CR)	<10	11–20	>20
1	X			X				X	X				X	
2		X		X			X		X				X	
3	X			X				X			X	X		
4	X			X			X			X			X	
5	X		X				X		X			X		
6		X		X				X	X				X	
7		X			X	X					X			X
8		X			X			X		X			X	
9	X			X				X			X		X	
10	X			X			X				X	X		
11	X			X				X			X	X		

**Table 2 healthcare-10-01742-t002:** Themes emerging from the interview data.

Themes	Sub-Themes	Representative Quote
Awareness of DBs.	i. Documentation ii. Education and training iii. Emotional intelligence	*“These behaviours are very common and, from a policy point of view, I believe that we need to document them so that we determine the prevalent ones so we find specific measures to address them. DBs in my opinion can only be adequately addressed if they are documented.”**(P#2; Female; PR)* *“The truth is radiography is more than technical proficiency as you have to deal with other health care professionals and erratic patients. In the education and training I think the part on psychology and patient management should emphasize a lot the interaction between other health care workers and patients. Students would then take the psychological and behavioural environment seriously.”**(P#2; Female; PR)* “… *it takes us to the point of emotional intelligence. You need to understand why a certain problem has occurred. Maybe that person has got a genuine problem, and maybe that person just wants to be difficult.”* *(P#4; Male; ACR)*
2. Willingness to address DBs.	i. Communication ii. Taking charge	*“As a hospital, there should be improved interprofessional communication to avoid conflicts among healthcare workers.”* *(P#1; Male; PR)* *“… the individual should take charge of their health because this is a wellness issue. Their wellbeing is obviously affected, their work environment will be affected and everything else will be affected including patient care. So the individuals themselves must take it upon themselves to seek ways … to address such challenges.”* *(P#11; Male; CR)*
3. Conflict resolution	i. Listen and understand ii. Substitution iii. External remedies	*“There are times when you have to resolve conflict, take people who are not working together well. Sit down with them and talk to them and make them understand why they are there and how a toxic work environment can affect the overall performance of the organisation. And when you resolve that conflict you realize that it won’t happen again.”* *(P#10; Male; CR)* *“I think that having time off from the environment where the abuse occurred can really help. The victim is supposed to be given some days off to go and recover at home in peace.”* *(P#1; Male; PR)* *“Most managers will say if I address this issue head-on…I will lose business if this guy leaves. So maybe if it gets addressed from a professional board-level …”* *(P# 10; Male; CR)*

## Data Availability

Not applicable.

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
