# Peer review of "Radiography Managers’ Perspectives on the Strategies to Mitigate Disruptive Behaviours: A Qualitative Exploratory Study"

_healthcare, 2022, doi:10.3390/healthcare10091742_

Round 1
Reviewer 1 Report
(1) You provided specific examples about what could be disruptive behavior. Just move those examples earlier in the manuscript. I was wondering what specific examples are those until I read them later in the manuscript.
(2) You said 4 open ended questions. Only 3 are stated, where is #4?
(3) Must explain more about NVivo for someone who does not know about the program (I know about it but not everyone does)
(4) See line #85 of your manuscript. It is confusing if you are interviewing managers (e.g. supervisors). Then line #86 you said RMS. Then line 394 you said radiography managers. It is confusing. Be consistent throughout the manuscript.
(5) Great you included a table with themes from the interview data. Perhaps a small table with demographics is helpful too.
(6) Good you did not identify the people you interviewed and included each in parenthesis.
(7) I love that portion about communication (line 351 and beyond). Communication is very important in these issues.
I enjoyed reading this manuscript! It is interesting and well done.
Author Response
Find my responses attached below

Reviewer 2 Report
I read a very interesting and original study entitled "RADIOGRAPHY MANAGERS' PERSPECTIVES ON THE STRATEGIES TO MITIGATE DISRUPTIVE BEHAVIOURS: A QUALITATIVE EXPLORATORY STUDY".
First of all, congratulations to the authors for the excellent manuscript they have submitted. I believe that this paper should be published after minor revision.
Here are my observations:
I would ask the authors not to use abbreviations in the abstract (specifically in line 14 abbreviation DB).
In the discussion it would be appropriate for the authors to make suggestions for future research.
I believe that a paragraph should be added at the end of the discussion with the limitations of the study.
